# Increasing Hydrostatic Pressure Impacts the Prokaryotic Diversity during *Emiliania huxleyi* Aggregates Degradation

Christian Tamburini [1,*], Marc Garel [1], Aude Barani [1], Dominique Boeuf [2], Patricia Bonin [1], Nagib Bhairy [1], Sophie Guasco [1], Stéphanie Jacquet [1], Frédéric A. C. Le Moigne [1], Christos Panagiotopoulos [1], Virginie Riou [1], Sandrine Veloso [1], Chiara Santinelli [3] and Fabrice Armougom [1,*]

1   Aix Marseille Univ., Universite de Toulon, CNRS, IRD, MIO UM 110, 13288 Marseille, France; marc.garel@univ-amu.fr (M.G.); aude.BARANI@univ-amu.fr (A.B.); patricia.bonin@mio.osupytheas.fr (P.B.); nagib.bhairy@mio.osupytheas.fr (N.B.); sophie.guasco@mio.osupytheas.fr (S.G.); stephanie.jacquet@mio.osupytheas.fr (S.J.); frederic.lemoigne@mio.osupytheas.fr (F.A.C.L.M.); christos.panagiotopoulos@mio.osupytheas.fr (C.P.); virginie_riou@hotmail.com (V.R.); sandrine.veloso@gmail.com (S.V.)
2   Sorbonne Université, CNRS, Laboratoire de Biodiversité et Biotechnologies Microbiennes, USR3579, Observatoire Océanologique, 66650 Banyuls-sur-Mer, France; boeuf.domi@gmail.com
3   C.N.R., Istituto di Biofisica, 56124 Pisa, Italy; chiara.santinelli@ibf.cnr.it
*   Correspondence: christian.tamburini@univ-amu.fr (C.T.); fabrice.armougom@mio.osupytheas.fr (F.A.)

**Abstract:** In the dark ocean, the balance between the heterotrophic carbon demand and the supply of sinking carbon through the biological carbon pump remains poorly constrained. In situ tracking of the dynamics of microbial degradation processes occurring on the gravitational sinking particles is still challenging. Our particle sinking simulator system (PASS) intends to mimic as closely as possible the in situ variations in pressure and temperature experienced by gravitational sinking particles. Here, we used the PASS to simultaneously track geochemical and microbial changes that occurred during the sinking through the mesopelagic zone of laboratory-grown *Emiliania huxleyi* aggregates amended by a natural microbial community sampled at 105 m depth in the North Atlantic Ocean. The impact of pressure on the prokaryotic degradation of POC and dissolution of *E. huxleyi*-derived calcite was not marked compared to atmospheric pressure. In contrast, using global $O_2$ consumption monitored in real-time inside the high-pressure bottles using planar optodes via a sapphire window, a reduction of respiration rate was recorded in surface-originated community assemblages under increasing pressure conditions. Moreover, using a 16S rRNA metabarcoding survey, we demonstrated a drastic difference in transcriptionally active prokaryotes associated with particles, incubated either at atmospheric pressure or under linearly increasing hydrostatic pressure conditions. The increase in hydrostatic pressure reduced both the phylogenetic diversity and the species richness. The incubation at atmospheric pressure, however, promoted an opportunistic community of "fast" degraders from the surface (*Saccharospirillaceae, Hyphomonadaceae,* and *Pseudoalteromonadaceae*), known to be associated with surface phytoplankton blooms. In contrast, the incubation under increasing pressure condition incubations revealed an increase in the particle colonizer families *Flavobacteriaceae* and *Rhodobacteraceae,* and also *Colwelliaceae,* which are known to be adapted to high hydrostatic pressure. Altogether, our results underline the need to perform biodegradation experiments of particles in conditions that mimic pressure and temperature encountered during their sinking along the water column to be ecologically relevant.

**Keywords:** biological carbon pump; carbon cycle; mesopelagic; mineral ballast; coccolithophorid; *Emiliania huxleyi*; prokaryotes; biodegradation; hydrostatic pressure

## 1. Introduction

The oceanic biological carbon pump, through a complex set of processes, removes a part of the atmospheric carbon dioxide ($CO_2$), fixed by photosynthesis in the euphotic

ocean, by storing it into the ocean's interior [1,2]. Hence, the dark ocean food webs and biogeochemical cycles are mainly fueled by gravitational sinking particles (the biological gravitational pump—BGP). Recently, the particle injection pumps have been proposed as an augmented BGP [3], aggregating additional transportation pathways such as water mixing process, eddy subduction, and diel or seasonal vertical migrations of fish or zooplankton. The mesopelagic or twilight zone (euphotic zone base to 1000 m depth) represents a key environment, in which gravitational sinking particles are transported and transformed [2,4], half of the flux loss being due to particle fragmentation and remineralization [5], driven by zooplankton [6–8] and by heterotrophic microorganisms [9,10]. While the twilight zone is at risk due to climate warming and potential human exploitations [11], little is known about the global composition, ecology, and functions of the microbiota inhabiting this vast ocean domain. The BGP functioning may be altered by global oceanic changes (increasing stratification, changes in upwelling and ice cover, acidification, lower oxygen, and higher temperature) according to future climate projections [12–14]. It is therefore highly relevant to estimate and understand how gravitational sinking particles are remineralized (into dissolved inorganic carbon) by heterotrophic prokaryotes through mesopelagic waters.

Using a marine snow catcher device, Baumas et al. [15] recently studied the link between diversity and activity of prokaryotes associated with freshly collected gravitational sinking particles at the Porcupine Abyssal Plain site (PAP site) in the Eastern North Atlantic Ocean. To complement these observations, we hereby present a laboratory biodegradation experiment carried out during the same field sampling campaign (DY032 cruise). In this paper, we evaluate the hydrostatic pressure impact, as a single abiotic forcing factor, on prokaryotic diversity and activity associated with gravitational sinking particles. Following the diatom-dominated spring bloom, the Eastern North Atlantic experiences some of the most extensive coccolithophore blooms on the planet [16]. Most of the deep North Atlantic sediments consist of rich calcareous materials composed of coccolithophores, such as *E. huxleyi,* implying a significant role for these organisms in efficient export [16,17]. The presence of "ballast" minerals (such as opal, calcium carbonate, or dust) has been proposed to preserve the organic matter from biotic degradation [18–23]. Prokaryotic-mediated diatom silica dissolution has been found to increase together with the degradation of organic matter in surface waters [24–26] and to decrease with decreasing temperature [27]. The addition of $CaCO_3$ to diatom aggregates was also observed to decrease silica dissolution by diminishing remineralization of the aggregates by zooplankton [28]. Laboratory studies confirm that calcifying cells such as coccolithophores can enhance particle density and sinking speed [29]. However, the link between $CaCO_3$ dissolution and particulate organic carbon (POC) degradation remains unclear, as there is still no evidence that $CaCO_3$ dissolution is microbiologically mediated [30] and referenced therein. Still, significant $CaCO_3$ dissolution occurs in the upper 1000 m of the water column, above the calcite or aragonite saturation horizon [31,32]. According to Sulpis et al. [33], 47% of the global $CaCO_3$ exported to 300 m depth might be dissolved in the water column.

In situ tracking of the dynamics at the particle level as they sink into the water column is still challenging. The particle sinking simulator (PASS) system is an experimental setup that accurately simulates the pressure increase (and the temperature decrease) that the prokaryotes associated with particles are experiencing during their sinking to depth [34]. Briefly, high-pressure bottles (HPBs) are used to incubate samples while pressure is linearly increased by means of a piloted pressure generator. Particles are maintained in suspension by rotating (semi-revolution) HPBs during the incubation in water baths mimicking temperature changes with depth. This unique experimental device has been previously used in a series of biodegradation experiments [34–36].

Based on the experimental design of Bidle and Azam [24,25] and Bidle et al. [27,37], the first pressurized biodegradation assay of Tamburini et al. [36] used diatom detritus (*Thalassiosira weisflogii*) as a source of POC and biogenic silica. In comparison with constant atmospheric pressure (ATM) conditions (as control), significantly lower aminopeptidase activity was measured under increasing pressure (HP), which, in turn, limited biogenic

silica dissolution to a simulated depth of 800 m [36]. Consistently with this first experiment, the degradation of freshly sampled in situ particles (mainly fecal pellets) was also slower under HP than at ATM conditions [34]. In contrast with the latter experiments, calcifying *E. huxleyi* aggregates were found to be more sensitive to degradation under HP than ATM conditions, partially compensated by an increase in aggregation under increasing hydrostatic pressure [35]. A stronger dissolution of particulate inorganic carbon (PIC) under HP has also been observed with diatom-biomineral (carbonate-kaolinite-smectite) aggregates, as well as a decrease in the rate of POC decay [38].

Although POC degradation rates decreased in the two sets of simulated sinking experiments [34,36], in agreement with [39], no drastic changes were detected in prokaryotic assemblages, as estimated by catalyzed reporter deposition fluorescence in situ hybridization methods (CARD-FISH). Gammaproteobacteria and Alphaproteobacteria dominated in both conditions, though the latter seemed to be affected by the increase in hydrostatic pressure [35]. Archaea were also identified in particle degradation, as was already observed in previous studies e.g., [9,25]. Changes in the finer taxonomic structure of the microbiota can, however, play a disproportionate role in the biogeochemical processes involved on particles since rare species may perform key environmental functions [40].

To finely monitor the phylogenetic changes occurring during particle sinking and to detail the effect of increasing hydrostatic pressure as the single abiotic forcing factor in this *Emiliania huxleyii*-aggregate biodegradation experiment, the present study describes the use of high-throughput sequencing of the transcript of the small ribosomal subunit (SSU, 16S rRNA) in particle-associated prokaryotes. We highlight the reduction in respiration rate of surface-originated community assemblages submitted to increasing hydrostatic pressure conditions as well as a drastic difference in the composition of transcriptionally active prokaryotes associated with *E. huxleyi*-derived particles comparing incubations performed at atmospheric and at increasing hydrostatic pressure.

## 2. Materials and Methods

### 2.1. SINking PArticles Simulation Experiments (SINPAS Experiments)

Incubation experiments were conducted in June–July 2015 onboard the RRS Discovery at the Porcupine Abyssal Plain Sustained Observatory (PAP-SO; 49°N, 16°W) site during the DY032 cruise. Figure 1 shows the complete design of the incubation experiments. On 28 June 2015, a total volume of 7 L of seawater sampled at 105 m depth was gently filtrated through pre-combusted GF/F filters (450 °C for 6 h, 47 mm in diameter) in order to obtain fresh natural <0.7 μm prokaryotic assemblages. Around 4 L of these natural prokaryotic assemblages were inoculated with aggregates made of an axenic culture of *E. huxleyi* TW1 strain, prepared following the protocol described in Riou et al. [35], to obtain a ca. 50 μM final POC concentration. This mix was volumetrically split into six fractions, using a peristaltic liquid dispenser (Jencons Scientific Ltd.,Leighton Buzzard, UK), pouring the fractions into pre-combusted glass bottles.

Duplicate samples (T0) were immediately sub-sampled for the biogeochemical parameters and diversity as described below. The four other aliquots were immediately transferred into high-pressure bottles (HPBs, 500 mL final volume) and fitted onto the particle sinking simulator (PASS) system [34]. Incubations were performed in duplicates for 6 days, either at atmospheric pressure (ATM) or continuously pressurized at a rate of of 1.5 MPa d$^{-1}$ (HP) corresponding to a simulated sinking rate of 150 m per day. Coccolithophore aggregates were kept in suspension by half-revolutions every minute of the 2-paired HPBs in 2 temperature-regulated water baths. After the 6-days incubation, T6-HP samples had reached 10.05 MPa, simulating the sinking of the aggregates (and their associated prokaryotic assemblages) to a depth of −1005 m, while T6-ATM samples had remained at atmospheric pressure. Both ATM and HP incubations were performed following a decrease in temperature (from 13 to 8.5 °C at the end of the incubation), agreeing with the temperature profile measured in situ between −105 and −1005 m, during the sampling period (Figure 1). HP samples were compared to ATM samples (in duplicate), in which the

only decrease in temperature was controlled in the same water baths simultaneously. For a complete description of the PASS system, see Tamburini et al. [34]), except for the online measurement of dissolved oxygen concentration described hereafter.

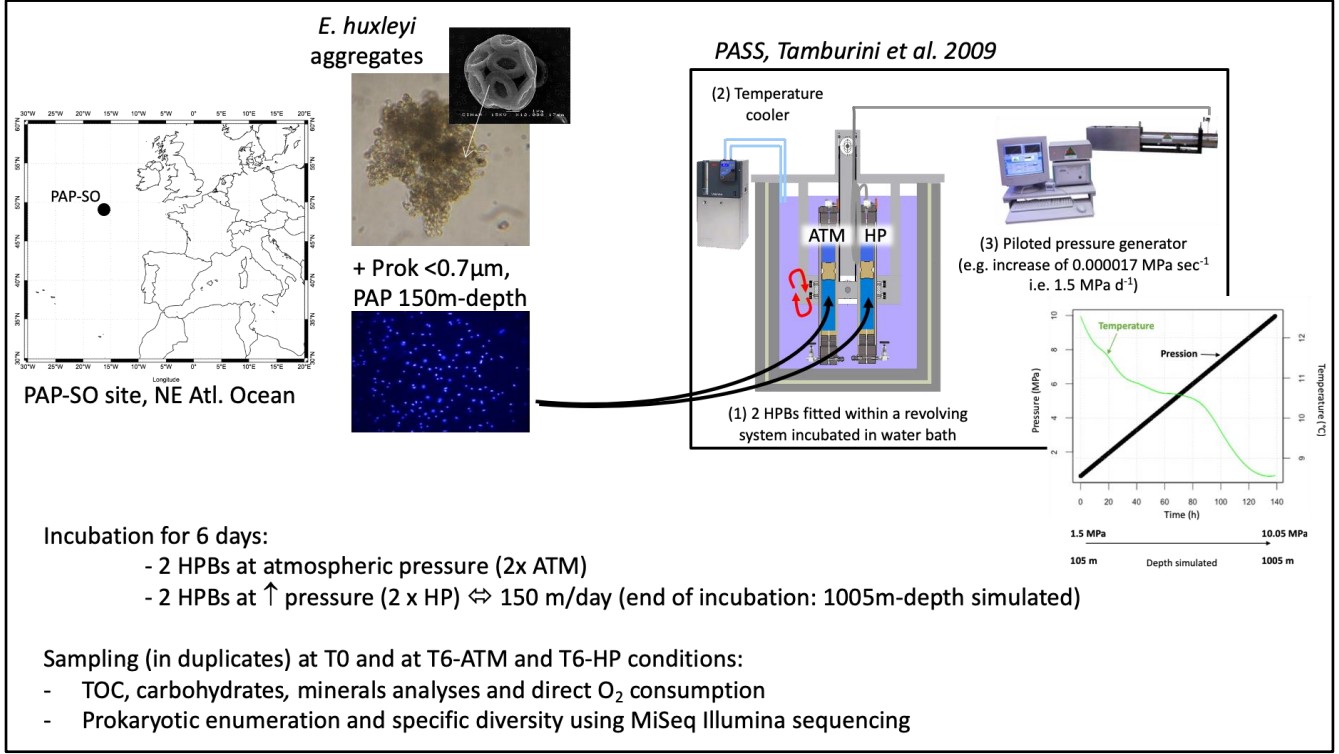

**Figure 1.** Schematic design of the sinking particles simulation (SINPAS) experiment. The objective of the present study was to investigate the effect of increasing hydrostatic pressure on the degradation of *Emiliania huxleyi* aggregates by NE Atlantic natural free-living prokaryotic assemblages collected at 105 m depth (PAP site). Using the particles sinking simulator system (PASS described in Tamburini et al. [34]), incubations were performed in duplicates for over 6 days under constant atmospheric pressure (ATM) or controlled increasing pressure conditions (HP) simulating a sinking rate of 150 m d$^{-1}$. Both ATM and HP conditions were submitted to a decrease in temperature mimicking profiles measured in situ by CTD (see plot temperature and pressure versus time). Global $O_2$ consumption was real-time monitored inside the high-pressure bottles using planar optodes via a sapphire window in both ATM and HP conditions. Three data sets were therefore obtained: initial time (T0), and after 6 days of incubation T6-HP and T6-ATM. The differences between T0 and T6-HP were due to the fate of *E. huxleyi* aggregates, the decrease in temperature, and the increase in hydrostatic pressure, those between T0 and T6-ATM were due to the fate of *E. huxleyi* aggregates and the decrease in temperature, and those between T6-HP and T6-ATM were due only to the increase in hydrostatic pressure.

At the end of the 6 days incubation, gentle depressurization was performed for the T6-HP samples. All the HPBs were opened and transferred in pre-combusted glass bottles prior to chemical and microbial sub-sampling.

### 2.2. Biogeochemical Analyses

All glassware was rinsed with 1 N HCl, Milli-Q water, and pre-combusted at 450 °C for 6 h prior to use. All plastic wares were rinsed with 1 N HCl and Milli-Q water and changed for each sample.

Samples (20 mL) for total organic carbon (TOC) were stored at −20 °C and analyzed at the Institute of Biophysics (IBF), CNR, Pisa, with a Shimadzu TOC-V$_{CSN}$ as described in Santinelli et al. [41]. Particulate carbohydrates (PCHO-C) were obtained by gently filtrating 30 mL throughout pre-combusted GF/F filters. Then, 10–15 mL of the filtrate were transferred with pre-combusted glass pipettes into pre-cleaned 50 mL falcon tubes for dissolved carbohydrates (DCHO-C) analysis. GF/F filters and falcon tubes were stored in

the dark at $-20\,°C$ until analysis. In the laboratory, samples were analyzed as described previously [42,43]. Total carbohydrates (TCHO-C) were obtained by summing PCHO-C and DCHO-C results.

Samples for particulate calcium (PCa) were obtained by gently filtrating 50 mL throughout 0.45 μm polycarbonate filters. Filters were rinsed with a few mL of Milli-Q water, dried ($50\,°C$), and stored in Petri dishes at room temperature for later analysis. In the laboratory, total digestion of filters was performed using a tri-acid mixture (0.5 mL $HF$/1.5 mL $HNO_3$/$HCl$ 1 mL; all Optima grade) in closed Teflon beakers overnight at $95\,°C$ in a clean pressurized room. After evaporation close to dryness, samples were re-dissolved into 10 mL of $HNO_3$ 2%. Solutions were analyzed for Ca and other elements of interest (i.e., Al, Sr, and Ba) by HR-ICP-MS (high-resolution inductively coupled plasma mass spectrometry; ELEMENT XR, ThermoFisher Scientific Inc., Waltham, MA, USA) according to Jacquet et al. [44]. The presence of remaining sea salt was checked by analyzing Na and the sea salt particulate Ca (and other elements analyzed). Their contribution was found to be negligible. The suspended matter was obtained by gently filtrating 50 mL through pre-combusted and pre-weighed GF/F filters, dried overnight, and reweighed.

Prokaryotic abundance was estimated by flow cytometry. Five milliliters of seawater were immediately fixed with glutaraldehyde (0.25% final concentration), freeze-trapped in liquid nitrogen, and stored at $-80\,°C$ until analysis. In the laboratory, the samples were thawed at room temperature and stained using SYBR Green II (Invitrogen®, Thermo Fisher Scientific, Waltham, MA, USA), then analyzed using a FACSCalibur (BD Biosciences®, San Jose, CA, USA) of the PRECYM flow cytometry platform (http://precym.mio.univ-amu.fr/, accessed on 17 September 2021), to enumerate heterotrophic prokaryotes with low nucleic acid content (LNA) and high nucleic acid content (HNA) as detailed in Girault et al. [45].

Dissolved oxygen concentration was real-time monitored every minute, inside one ATM high-pressure bottle and one HP high-pressure bottle using a non-invasive planar optode method (optical oxygen-sensor and OXY-10 mini device, Presens GmbH®, Regensburg, Germany) following a protocol and design presented in Garel et al. [46]. Raw $O_2$-data measured using planar $O_2$ sensing foil under increasing pressure conditions were corrected by post-treatment to take into account at each measurement point the hydrostatic pressure using the calibration equation of McNeil and D'Asaro [47]. Temperature decrease was also corrected for both incubations. Respiration rates were estimated, in both conditions, by linear regression.

### 2.3. 16S rRNA Extraction, PCR Amplification, and Sequencing

Fifty milliliters of initial time duplicates (T0) and of final-time duplicates after incubation at atmospheric pressure (T6-ATM) or at increasing pressure conditions (T6-HP) were filtered through 0.2 μm-pore-size filters (hydrophilic polyethersulfone membrane 47 mm, GPWP04700 Millipore Corp., Burlington, Massachusetts, USA). Filters were flash-frozen in liquid nitrogen until further processing. Nucleic acids extractions were performed in duplicate, after gentle thawing on ice, with TE-Lysis buffer (20 mM Tris, 25 mM EDTA, 1 μg μL$^{-1}$ Lysozyme) containing 10% SDS, and phenol:chloroform:isoamyl alcohol (1:1:1, pH 6). Resulting RNA samples were treated with TurboDNase™ (Ambion®, ThermoFisher Scientific, Waltham, MA, USA) until any trace of DNA remains and then reverse transcribed into cDNA by RT-PCR using SuperScript® IV Reverse Transcriptase with random primers (Life Technologies, ThermoFisher Scientific, Waltham, MA, USA) following the manufacturer's recommendations. The hypervariable regions V4 of the SSU were amplified with universal primer sets [48], 515F-Y (5′-GTGYCAGCMGCCGCGGTAA-3′, [49]) and 806RB (5′-GGACTACNVGGGTWTCTAAT-3′, [50]), using 2.5 U/50 μL TaKaRa PrimeSTAR® GXL DNA polymerase (OZYME). Amplicons were sequenced by MiSeq Illumina (paired-end, $2 \times 250$ pb) at the Genotoul platform (https://get.genotoul.fr/en/ accessed on 17 September 2021). The raw data are available under the NCBI SRA project PRJNA731017.

### 2.4. Analysis of the 16S rRNA-Based Community

The cDNA-based community is referred to as the "16S rRNA-based community". The 16S rRNA-based community is interpreted as the "actively transcribing prokaryotes" since 16S rRNA is not a direct indicator of activity but rather protein synthesis potential [51]. Hereafter, "active" will thus refer to "actively transcribing prokaryotes". Raw sequencing reads (paired-end, $2 \times 250$ bp) were analyzed using DADA2 v1.8, a model-based approach for correcting amplicon sequencing errors [52]. After inspection of quality read profiles, the 16S rRNA paired-end reads were quality-trimmed (maxEE = c(2,3)), and only reads >150 bp were retained. The paired-end reads were then dereplicated, denoized (DADA2 error correction model), assembled, and chimeras were discarded (see "Results and Discussion" section). The high-quality and denoizing sequences obtained are amplicon sequence variants (ASVs). The taxonomic assignment of ASVs was performed using the SILVA_132 database [53], and 100% of sequence identity is required for species level. Finally, after sub-sampling normalization, $\alpha$- and $\beta$-diversity were characterized by R packages including phyloseq v1.36 [54], vegan v2.5–7 [55], Microbiome v1.14 [56] and ComplexHeatmap v2.7 [57]. Z-scores are abundance data normalized and standardized. The Z-scores are given in standard deviation (SD) to the population mean.

The global richness and the $\alpha$-diversity indices (Observed, ASV, Shannon, Simpson, and PD_faith) were calculated from phyloseq v1.36 and abdiv v0.2 [58]. To decide whether the data are homogeneous or heterogeneous, respectively (and thus more suitable for linear or unimodal ordination methods, respectively), we calculated detrended correspondence analysis (DCA) with vegan and checked the length of the first ordination axis in units of SD. The length of the first axis was 2.92 SD units, which means that linear ordination methods such as principal component analysis (PCA) are suitable [59]. PCA was computed with prcomp from the stats R package [60]. The PCA was performed at the family level because genus level indicated a large fraction of "not assigned" sequences (30% in HP and T0). To avoid the double-zeros problem [61], low abundant families were discarded (<0.5%). Only the most contributing families were displayed using the significant contribution scores from fviz_contrib of factoextra from stats v3.6.2 R package. The environmental variables were then fitted onto PCA ordination using envfit function of vegan by calculating multiple regression. Coordinates onto PCA ordination, $R^2$ scores, and *p*-values for each environmental variable were obtained from envfit from stats R package. Differences between T6-HP and T6-ATM conditions were analyzed by the non-parametric pairwise Mann–Whitney test on raw data (i.e., individual replicate values) due to the low number of replicates.

Finally, the $\beta$-Nearest-Taxon-Index ($\beta$NTI) [62,63] was used to quantify the phylogenetic turnover of communities between groups (T6-ATM, T6-HP, and T0), estimated using the picante R package [64]. The $\beta$NTI quantifies the magnitude and direction of deviation between an observed $\beta$MNTD (Mean-Nearest-Taxon-Distance) value and the mean of the null $\beta$MNTD distribution in SD units. The $\beta$NTI values ranging from $-2$ to $2$ indicate the dominance of stochastic processes, while $|\beta NTI| > 2$ indicates the dominance of deterministic processes [65].

## 3. Results and Discussion

### 3.1. Biogeochemistry of E. huxleyi Particles in Sinking Particle Simulation Experiments

TOC-normalized concentration of total carbohydrates (TCHO-C/TOC) significantly decreased (Mann–Whitney pairwise test, $p = 0.05$) under both conditions between the initial time (T0 = 5.6% $\pm$ 0.7%) and the 6-days incubation (T6-HP = 1.2% $\pm$ 0.1%; T6-ATM = 2.0% $\pm$ 0.5%; Figure 2A). Carbohydrates were equally degraded in both experimental conditions since no significant differences between ATM and HP conditions were noticeable (Mann–Whitney pairwise test, $p > 0.05$). Previous investigations with fecal pellet particles showed that particulate carbohydrate, chloropigments, and transparent exopolymer particles (TEP) concentrations decreased more slowly under HP than ATM conditions [34]. Increasing hydrostatic pressure conditions also influenced the efficiency of silica dissolution, as

well as prokaryotic activity (aminopeptidase), during another sinking particle simulation experiment with diatom (*Thalassiosira weissflogii*) aggregates [36]. The latter results agree with a recent study of Liu et al. [66], which showed, using a $^{13}$C-amended diatom culture (*T. weissflogii*), that degradation rates by the particulate-attached microbiote were markedly slowed down by increasing pressure.

In the present study, *E. huxleyi* coccoliths were partially dissolved, as the ratio of particulate calcium to suspended matter was twice lower after 6 days of incubation under both conditions (T6-ATM, T6-HP) compared to T0 (PCa/Susp., Figure 2A). However, as for the TCHO-C/TOC ratio, no significant difference in the particulate calcium/suspended mater ratio (PCa/Susp., Figure 2A) between T6-ATM and T6-HP conditions was noticeable. Another study using diatom-mineral aggregates (carbonate-kaolinite-smectite) found that particulate inorganic carbon dissolution was enhanced under HP conditions [38]. Using coccolithophorid aggregates (*E. huxleyi*), we previously hypothesized that increased hydro-static pressure may cause coccoliths to dissolve more rapidly than in surface waters [35]. Concurrently, Dong et al. [67] highlighted an in vitro pressure-dependent enhancement of PCa dissolution rates by a factor of 2 to 4 at 7 MPa (equivalent to a depth of 700 m) compared to dissolution at 0.1 MPa (atmospheric pressure).

Hence, the impact of increasing hydrostatic pressure on the degradation processes occurring in the mesopelagic zone could be dependent on the quality and source of carbon within the aggregates, as well as on the aggregate types themselves (Table 1). When increasing pressure, fecal pellets as well as diatom aggregates would better protect the access to organic carbon available for microbial degradation than particles composed of *E. huxleyi* coccoliths. Such variations in degradation efficiencies could also indicate differences in microbial communities associated with different types of aggregates or depths.

### 3.2. Prokaryotic Activity Associated with E. huxleyi Particles in Sinking Particles Simulation Experiments

The dynamics of total prokaryotic abundance were monitored by flow cytometry in HP and ATM conditions (Figure 2B). The abundance of LNA cells remained constant, while HNA cells increased between the beginning and the end of the experiment. HNA cells would represent the active fraction of prokaryotes, while LNA cells would represent the dormancy fraction of prokaryotes [68]. This increase was particularly marked under ATM conditions, which resulted in a significant difference (Mann–Whitney test, *p*-value = 0.03) in total cell abundance between HP and ATM incubation conditions (Figure 2B). However, during this experiment, the growth rate of total prokaryotes was lower compared to our previous studies on similar aggregates [35]. This was probably caused by colder incubation temperatures. In this work, the temperature decreased from 13 to 8.5 °C, mimicking the gradient encountered between 150 and 1000 m depth in the NE Atlantic Ocean, whereas in the previous experiments, the temperature stayed above 13 °C, mimicking the temperature observed in the mesopelagic waters of the NW Mediterranean Sea [34–36]. In Turley [39], while increasing pressure had no significant influence on cell numbers, it showed an impact on DNA and protein syntheses. Increasing pressure plays a role in the activity of the surface prokaryotic community, diminishing their activity while they sink in the water column, as previously mentioned in Tamburini et al. [36,69].

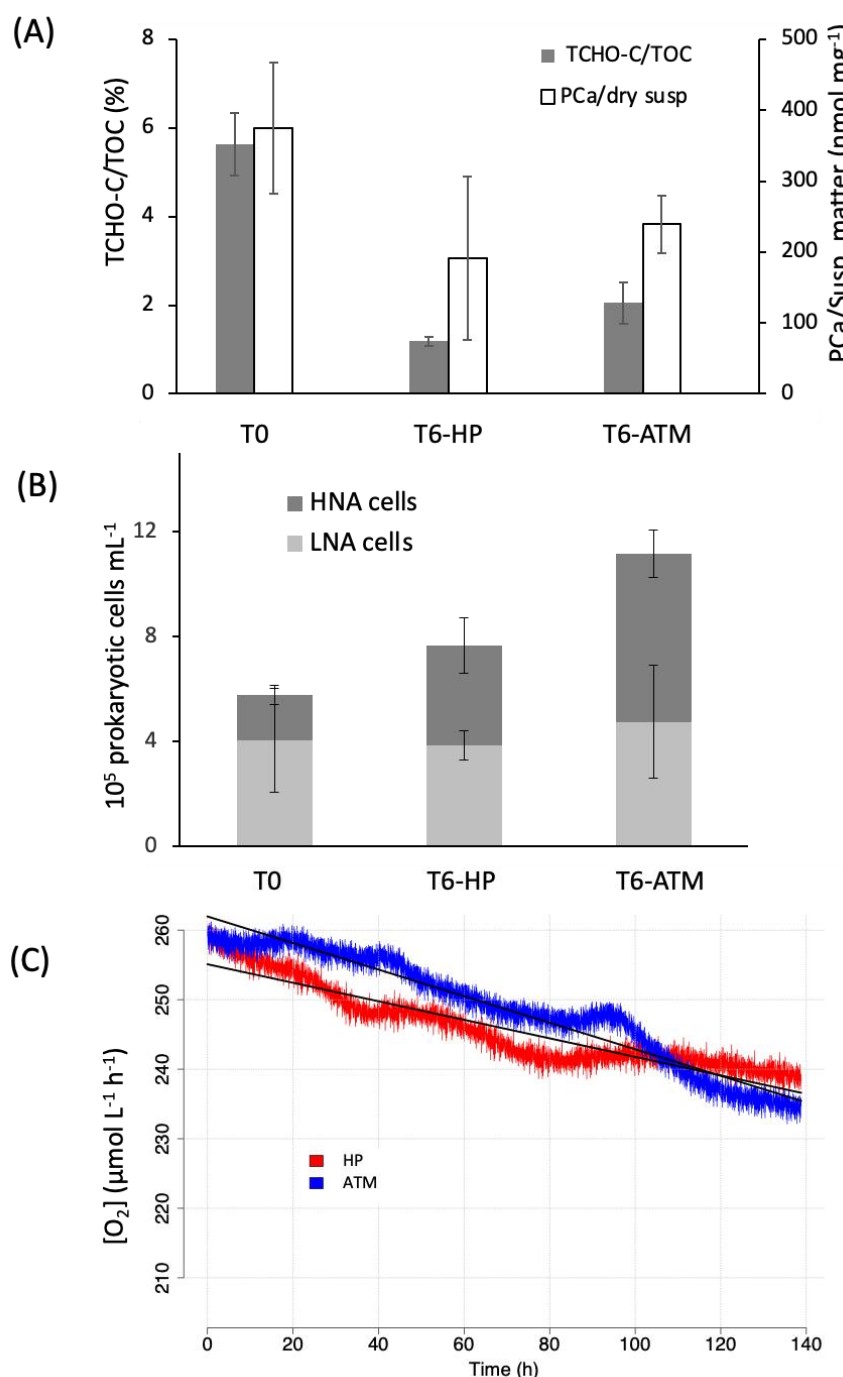

**Figure 2.** (**A**) Ratio of total carbohydrates (TCHO-C) versus total organic carbon (TOC) (left axis) and between particulate calcium and suspended matter (PCa/Susp.) expressed in nmol mg$^{-1}$ (right axis) initially (T0) and after 6 days of incubation under increasing pressure (T6-HP) or atmospheric pressure (T6-ATM) conditions. Error bars display the standard deviation of the duplicates in both incubation conditions. (**B**) Prokaryotic abundance initially (T0) and after 6 days of incubation at HP and ATM conditions. LNA (light gray) are cells with low nucleic acid content, HNA (dark gray) are cells with high nucleic acid content. Total cells correspond to the sum of LNA and HNA cells. Error bars display the standard deviation of the duplicates in both incubation conditions. (**C**) Real-time measurements of the total O$_2$ consumption under conditions of increasing pressure (HP, red line) and atmospheric conditions (ATM, blue line). Black lines correspond to the linear regression used to calculate oxygen consumption rates; HP: $-0.13 \pm 0.0005$ μM O$_2$ L$^{-1}$ h$^{-1}$, R$^2$ = 0.90; ATM: $-0.19 \pm 0.0005$ μM O$_2$ L$^{-1}$ h$^{-1}$, R$^2$ = 0.96.

**Table 1.** Main results from literature reporting similar experiments performed with the surface prokaryotic community. ND: not determined.

| References | Aggregates | Pressure Simulated | Depth Simulated | Temperature | Sinking Rate | Dissolved $O_2$ Measurement | Microbial Diversity | Main Results |
|---|---|---|---|---|---|---|---|---|
| [36] | Aggregate-forming diatom *T. wessflogii* | 2–14 MPa | 200–1400 m | 13 °C | Pressure increase of 1.5 MPa each day | ND | CARD-FISH | Increasing pressure decreased silica dissolution and aminopeptidase activity (relatively to constant ATM conditions) and increased Bacteroidetes abundance |
| [34] | Fecal pellets | 2–15 MPa | 200–1500 m | 13–13.4 °C | Continuously pressurized at a rate of 2 MPa d$^{-1}$ | ND | CARD-FISH | Particulate carbohydrate, chloropigments and TEP decreased more slowly under HP than ATM condition |
| [35] | Aggregate-forming coccolithophorid *E. huxleyii* | 1–17 MPa | 100–1700 m | 13 °C | Continuously pressurized at a rate of 1.5 MPa d$^{-1}$ | ND | CARD-FISH | Increasing pressure enhanced dissolution of calcite and particle aggregation (relatively to constant ATM conditions), and decreased α-Proteobacteria abundance |
| [67] | Synthetic inorganic Ca$^{13}$CO$_3$ (calcite) | 25 MPa | 2500 m | 21 °C | ND | ND | ND | Increasing pressure promoted calcite dissolution at 7 MPa |
| [38] | Culture of *Thalassiosira weissflogii* | 10 MPa | 1000 m | ND | Continuously pressurized at a rate of 3 MPa d$^{-1}$ | ND | ND | Increasing pressure enhanced particulate inorganic carbon dissolution |
| [70] | No aggregates | 40 MPa | 4000 m | 20.5 °C | Pressure increase of 10 MPa each day | Yes | FISH * | Increasing pressure inhibited growth of surface-originated bacteria and decreased α-Proteobacteria et Bacteroidetes abundance |
| [71] | Aggregate-forming diatom *Nannochloropsis* and *Tetraselmis* algae | 30 MPa | 3000 m | 4 °C | Pressure increase of 2.5 MPa every 15 min | Yes | ND | Increasing pressure promoted particulate inorganic carbon dissolution |
| [72] | Diatom-bacteria aggregates | 10 to 100 MPa | 100–10,000 m | 3 °C | Pressure increase or decrease of 10 MPa during 15–20 min | Yes | ND | Increasing pressure inhibited respiration on surface prokaryotic assemblage |
| [72] | Aggregate-forming diatom *Skeletonema marinoi* | 10 to 100 MPa | 100–10,000 m | 3 °C | Pressure increase or decrease of 10 MPa during 15–20 min | Yes | ND | Increasing pressure inhibited respiration on surface prokaryotic assemblage |
| [This study] | Aggregate-forming coccolithophorid *E. huxleyii* | 1.05–10.05 MPa | 105–1005 m | 13–8.5 °C | Continuously pressurized at a rate of 1.5 MPa d$^{-1}$ | Yes | Metabarcoding | Increasing pressure inhibited respiration on surface prokaryotic assemblage, reduced both the phylogenetic diversity and the species richness (specifically Bacteroidetes) |

* Microbial community was artificial, composed with 5 different strains of Bacteria.

To assess changes in prokaryotic activity with increasing pressure, we monitored real-time $O_2$ consumption for both HP and ATM conditions (Figure 2C). This indicates rates of prokaryotic aerobic respiration activity. As far as we know, very few studies investigated the impact of pressure on particle respiration activity by direct measurements of $O_2$ consumption [70–72]. All of these investigations showed that no significant differences occurred in the first simulated 1000 m of the water column. In our incubation experiments, $O_2$ consumption rates were slightly higher at ATM compared to HP conditions ($-0.13 \pm 0.0005$ versus $-0.19 \pm 0.0005$ µM h$^{-1}$, for HP and ATM conditions, respectively, t-test comparison regression slopes *p*-value = 0.001). These respiration rates are in line with Stief et al. [72], showing inhibition of respiration when surface prokaryotic assemblages were exposed to pressure levels of 10–50 and 60–100 MPa. Under ATM conditions, more competitive opportunistic taxa could grow by increasing the respiration rate unrealistically, as attested by the diversity found (see thereafter). A previous study has shown that a change of environmental conditions leads to a change in microbial community composition, privileging opportunistic microorganisms at atmospheric pressure [46]. In the following section, we investigate how prokaryotic species richness evolved during our incubations.

### 3.3. Effect of the Increasing Pressure on the Active Prokaryotic Communities

To further evaluate the effect of increasing pressure on the active community composition (term used hereafter referring to "actively transcribing prokaryotes", see Methods section), a 16S rRNA metabarcoding survey was performed during the degradation experiment. A total of 445,620 raw reads were generated, ranging from 59,218 to 98,599, depending on the sample. After the quality trimming process, about 69%–76% of reads were retained (Table 2). The global species richness (observed ASV counts) ranged from 54 to 106 ASVs, depending on the sample (Table 2). Overall, the T6-HP samples showed a lower global richness ($65 \pm 11$) than those of the T6-ATM ($94 \pm 4$) and the T0 ones ($101 \pm 5$). However, regarding the Shannon and Simpson diversity indices (Table 2), which consider the evenness, a small decrease in the α-diversity level is observed under ATM conditions compared to either T0 or T6-HP samples. This observation results from a more pronounced imbalance of the evenness within the microbial community under ATM conditions, a direct consequence of the presence of highly dominant ASVs (in relative abundance). Moreover, estimating the phylogenetic diversity indices (PD) [73] indicates higher phylogenetic diversity within T0 (9.46) and T6-ATM (8.63) compared to T6-HP groups (5.86). Altogether, these findings indicate that the degradation experiment under increasing pressure reduced both species richness and phylogenetic diversity independently of the degradation process of organic matter. Indeed, no significant difference was observed for the TCHO-C/TOC ratio, as well as for the particulate calcium/suspended matter ratio (PCA/Susp. Figure 2A) between ATM and HP conditions.

**Table 2.** Data quality trimming and alpha diversity indices. The columns headings of Table 2 indicate the number of reads retained after each processing step of DADA2 from the initial raw data (Input). "Filtered" column corresponds to the removal of bad quality reads, "Denoised" corresponds to the correction of sequencing errors, "Merged" are assembled reads and "Non chimeric" are not artefact sequences. The "Final retained" column is the read proportion conserved for downstream analyses. The "global richness" (which corresponds to the number of observed amplicon sequence variants—ASVs), Shannon, Chao1, and PD (phylogenetic diversity) columns provide alpha diversity indices calculated for each sample.

| Conditions | Samples | Input | Filtered | Denoised | Merged | Non-Chimeric | Final Retained (%) | Global Richness | Shannon | Chao1 | PD |
|---|---|---|---|---|---|---|---|---|---|---|---|
| T0 | T01 | 59,218 | 43,696 | 43,524 | 42,323 | 42,242 | 71 | 106 | 4.13 | 106 | 10.47 |
| | T02 | 71,661 | 54,986 | 54,827 | 53,562 | 53,412 | 75 | 96 | 3.86 | 96 | 8.45 |
| T6-ATM | ATM1 | 50,037 | 36,087 | 35,891 | 34,808 | 34,746 | 69 | 90 | 2.55 | 104 | 8.80 |
| | ATM2 | 98,599 | 72,212 | 71,853 | 70,168 | 69,053 | 70 | 98 | 2.14 | 99 | 9.21 |
| T6-HP | HP1 | 94,669 | 72,866 | 72,701 | 71,523 | 71,500 | 76 | 76 | 3.83 | 76 | 6.20 |
| | HP2 | 71,436 | 53,275 | 53,177 | 52,386 | 52,386 | 73 | 54 | 3.50 | 54 | 5.52 |

The overall taxonomic community composition of samples was spread across 21 phyla (Figure 3). The bacterial phylum Proteobacteria dominated most of the active community in all the conditions, accounting for 68.7%, 93.9%, and 60.2% of the reads (on average) within T0, ATM, and HP groups, respectively. The Bacteroidetes, Firmicutes, and Actinobacteria were the other major bacterial phyla, accounting (on average) for 10.2%, 1.4%, and 18.1% for T0, 7.4%, 3.3%, and 5.8% for T6-ATM and 7.7%, 0.5%, and 6.7% for T6-HP samples. All the other bacterial phylum were represented by less than 2% in all conditions. The archaeal community, which encompasses the Euryarchaeota, Nanoarchaeota, Crenarchaeota, and Thaumarchaeota phyla, represented 1.7%, 0.1%, and 5.5% (on average) of the reads within T0, T6-ATM, and T6-HP samples, respectively. Finally, specific community structures emerged according to the experimental conditions (Figure 3). In that respect, the T0 samples (initial time) harbored a community profile dominated by Proteobacteria (68.7%) associated with similar abundance levels of Firmicutes (7.4%), Bacteroidetes (10.2%), and Actinobacteria (7.7%). In contrast, the T6-ATM community profile consisted of almost exclusively Proteobacteria (93.9%), with the lowest Firmicutes (3.3%), Bacteroidetes (1.4%), and Actinobacteria (0.5%) abundance levels, compared to T0 and T6-HP samples. Lastly, the T6-HP community profile showed the lowest Proteobacteria level (60.2%) to the benefit of Bacteroidetes (18.1%) and similar levels of Actinobacteria (6.7%) and Firmicutes (5.8%) to those of T0. These results emphasize the variations of dominant phyla within the active community exposed to different experimental conditions. Abundance variations occurred between T6-ATM and T6-HP samples after a 6-day incubation period, starting from a common (T0) initial prokaryotic community (itself exhibiting a certain level of variability between duplicates).

Bacteroidetes appeared to be positively affected by increasing pressure (highest abundance level for T6-HP samples), which may have been stimulated by the supply of fresh, complex organic matter released by particles, as already reported elsewhere [74,75]. Using a similar 16S rRNA metabarcoding, Liu et al. [66] recently found, from a $^{13}$C-amended diatom culture degradation experiment performed at pressures corresponding to 75 m, 2000 m and 4000 m depth, that Gammaproteobacteria, Alphaproteobacteria, and Bacteroidetes were the most active phyla in POC degradation. As already observed in previous similar biodegradation experiments using catalyzed reporter deposition fluorescence in situ hybridization (CARD-FISH) for phylogenetic distinctions [34–36], a very low abundance of Archaea (<3%, initially) was present after 6 days of incubation (except in one of the duplicate-sampled at T6-HP where Euryarchaeota reached 10.5% of the whole community).

As β-diversity analysis, a hierarchical clustering (based on Bray–Curtis dissimilarity of community composition) combined to a Z-score heatmap of the 20 most active families (relative abundance > 0.5%) showed the reunification of samples by experimental conditions (T0, T6-ATM or T6-HP) (Figure 4). Accordingly, Figure 4 exhibits a strong divergence between the abundance family profiles of T6-ATM, T0, and T6-HP groups. For the T6-ATM samples, the representative family pattern is indicated by a relative overabundance of the opportunistic *Pseudoalteromonadaceae* (+1.12 ± 0.002, Z-score ± SD), *Hyphomonadaceae* (+2.68 ± 0.24, Z-score ± SD) and *Saccharospirillaceae* (+2.71 ± 0.28, Z-score ± SD) compared to the mean. In contrast, each of these families is slightly underrepresented within T0 and T6-HP samples (Z-score ≤ 0.5). These three opportunistic families belong to the Gammaproteobacteria, which account for 85% of the total Proteobacteria fraction (93.9%) at ATM conditions (Figure 3).

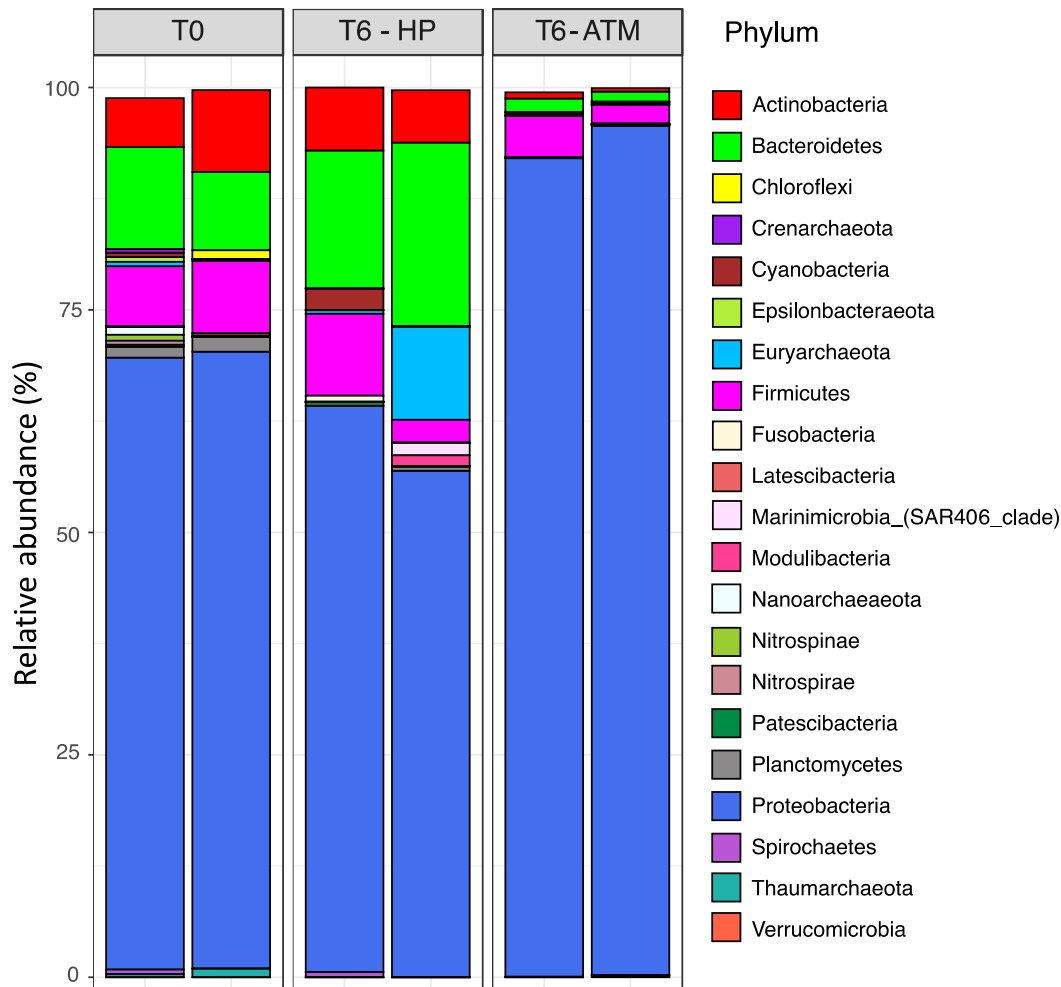

**Figure 3.** Relative abundance (%) of the most actively transcribed prokaryotes (>0.01% in average) initially (T0) and after 6 days of incubation under increasing pressure (T6-HP) and atmospheric pressure (T6-ATM) conditions.

At atmospheric pressure, the genus *Oceanicaulis* of the family *Hyphomonadaceae* is predominant (28% of all genera and represented by only one ASV), whereas it reaches only 2.6% and 0.6% within T0 and T6-HP, respectively. The genus *Oceanicaulis* has been associated with *E. huxleyii* blooms and is one of the main consumers of alkenones produced by *E. huxleyii* [76]. Strains of *Pseudoalteromonadaceae* were also exacerbated during a recent particles degradation experiment performed at atmospheric conditions [77]. This family is considered as "fast" degraders (copiotrophic) and primary colonizers on particles, partially due to their ability to swim [77].

The bacterial families mainly responsible for the T6-HP pattern were the *Flavobacteriaceae* (Z-score = +2.94 SD ± 0.2), *Rhodobacteraceae* (Z-score = +2.03 SD ± 0.30), and to a lesser extent, *Colwelliaceae*. In our incubation experiments, *Flavobacteriaceae* (Bacteroidetes) and *Rhodobacteraceae* (Alphaproteobacteria), each representing 7.1% and 7.4% at T0, reached 16.3% and 12.4% at the end of incubation under HP condition, respectively.

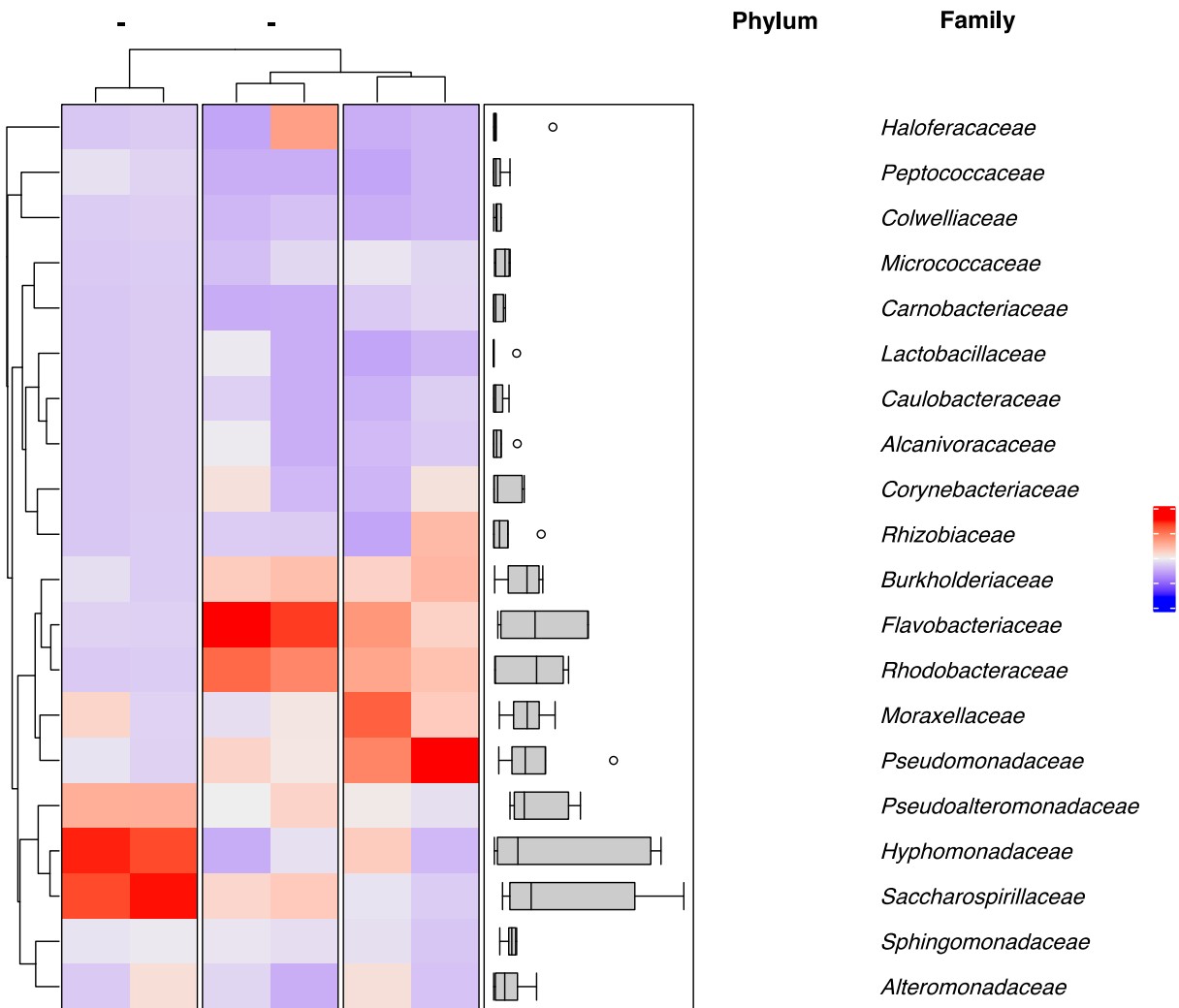

**Figure 4.** Heatmap visualizing the Z-score distribution from the relative abundance of the dominant families (>0.5% in average) initially (T0) and after 6 days of incubation under increasing pressure (T6-HP) and atmospheric pressure (T6 ATM) conditions. Z-score is the number of standard deviations separating a raw score from the mean. For each sample, we calculated how far from the mean (of a family relative abundance considering all samples) is the family abundance score of the sample. A positive Z-score indicates the raw score is higher than the mean (red color), and a negative Z-score reveals the raw score is below the mean (blue color). The higher the Z-score, the more overrepresented the family is for the sample compared to the mean. The dendrogram clusters were calculated with the UPGMA method (unweighted pair group method with arithmetic mean) according to the Bray–Curtis similarity index. The boxplots in the right panel show the distribution of family relative abundances (in %) for all samples.

### 3.4. Factors Driving the Active Prokaryotic Communities

Figure 5 is a principal component analysis (PCA) performed at the family level on which the environmental variables were fitted (see Methods section). It identifies the same taxonomic family patterns as the major compositional factors (of the global community) that govern the distinction of samples, as the Z-score heatmap (Figure 4). The PC1 axis (46.2% of total variance) mainly drives the position of T6-ATM samples by a positive gradient of *Hyphomonadaceae, Pseudoalteromonadaceae, Saccharospirillaceae,* but none of the environmental factors measured is linked to PC1 since the best candidate "HNA" shows a weak correlation ($R^2 = 0.67$) and no significance ($p$-value = 0.21) with the PC1 axis. The PC2 axis (22.7% of total variance) mainly leads to the discrimination of T6-HP samples from others by another positive gradient of *Flavobacteriaceae*, *Rhodobacteraceae*, and *Colwelliaceae*.

Interestingly, the multiple regression of environmental variables on the PCA ordination shows a strong correlation of the "Pressure" factor with the PC2 axis ($R^2 = 0.97$) and close to the significance threshold (*p*-value = 0.067, probably due to the small number of samples—duplicates). This trend links the positive gradient of *Flavobacteriaceae*, *Rhodobacteraceae,* and *Colwelliaceae* to the increased pressure, suggesting that increased pressure is one of the potential factors that may shape the specific family pattern of HP communities relative to those of the initial time—T0 (Figure 5).

Phylogenetic community turnover (at the ASV level) between sample groups (T0 vs. T6-ATM and T0 vs. T6-HP) was assessed by the β Nearest-Taxon-Index (βNTI). This index helps to understand the forces that influence the community composition, such as deterministic selection by environmental factors and/or ecologically neutral (stochastic) processes [63,65]. We thereby calculated βNTI to determine the stochastic/deterministic balance in the community assembly observed within each HP and ATM sample after a 6-day degradation experiment by increasing pressure (T6-HP) or not (T6-ATM) starting with the same initial community (T0). The βNTI score between "T0 vs. T6-HP" and "T0 vs. T6-ATM" is +2.57 and +0.02, respectively. A |βNTI| < 2 for "T0 vs. T6-ATM" pairwise community comparison excluded selection as the dominant process. In our experiment, the environmental factors are controlled in a closed system, as factors that change over time do not impose selection in ATM, neutral processes are expected to be dominant, here opportunistic bacteria were favored. Hence, the three strong competitors (having ecological traits of r-strategy) gammaproteobacterial families dominated at ATM conditions (*Pseudoalteromonadaceae*, *Hyphomonadaceae,* and *Saccharospirillaceae*). The dynamics of surface community incubated in ATM conditions can be compared to what occurs to deep-sea communities incubated at atmospheric pressure after decompression, in which distortion of the diversity occurred in comparison with samples maintained in in situ high-pressure conditions [46]. Consequently, biodegradation experiments simulating particles sinking along the water column but carried out at atmospheric pressure could be stated as ecologically biased.

In contrast, the "T0 vs. T6-HP" pairwise community comparison has a βNTI > 2, suggesting selection as the dominant process. Starting from the initial time (T0), the incubation experiments were performed by changing only one factor: the hydrostatic pressure gradient. The increase in hydrostatic pressure gradually enhanced the strength of selection (the selective pressure) and could have led to the exclusion of taxa [78], as suggested from the alpha diversity results (richness and phylogenetic diversity reduction).

Although this work deals with a limited sample number (due to technical constraints), our results clearly suggest a selection process driven by increasing hydrostatic pressure conditions (Figure 5). This environmental factor, that microbes experienced while they sink with gravitational sinking particles, leads to a reduction in both species richness (Table 2) and phylogenetic diversity (e.g., 35 and 34 families present in T6-ATM and T0, respectively, were no longer in T6-HP), and reshapes the community structure (shifts in phyla and family abundance, Figures 4 and 5). Thus, most interestingly, the results of these laboratory sinking particles simulation experiments agree with those obtained on freshly recovered particles samples using a marine snow catcher [15].

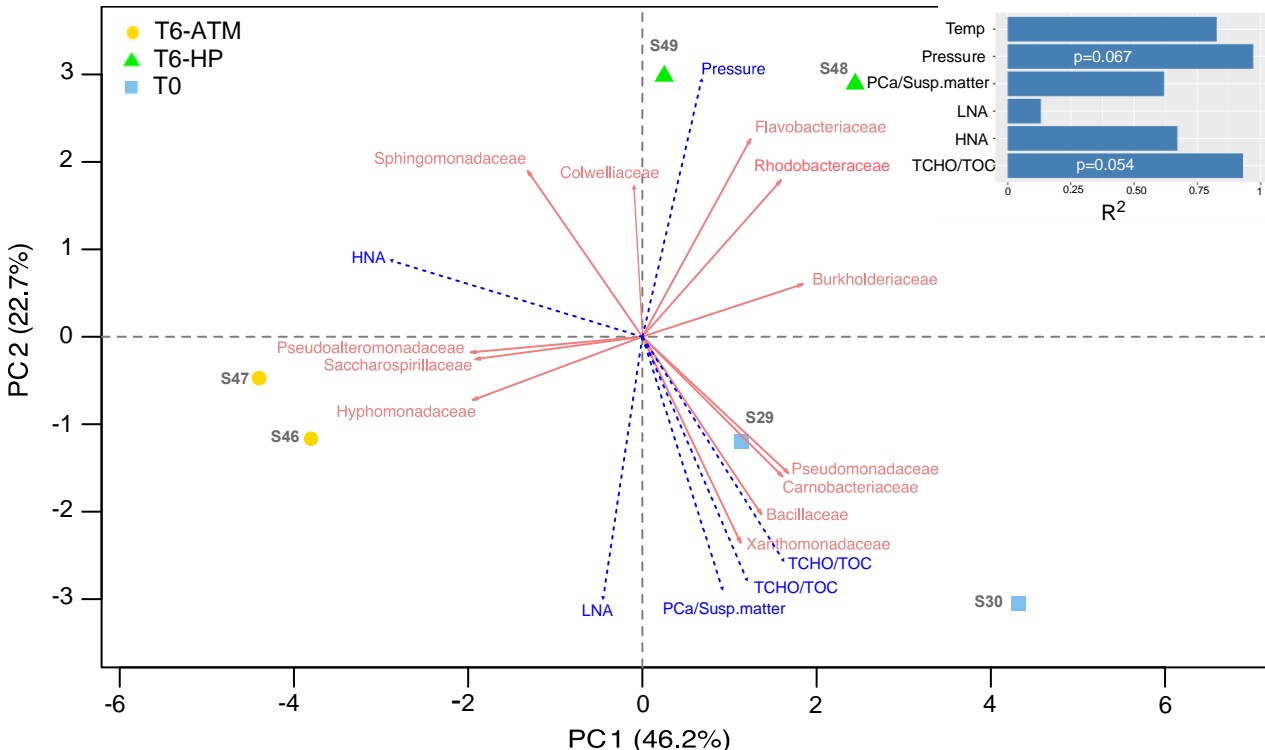

**Figure 5.** Principal component analysis (PCA) biplot of samples and the most contributing taxonomic families (based on fviz_contrib function of factoextra from stats v3.6.2 R package). Samples are indicated as yellow circles (T6-ATM), green triangles (T6-HP), or blue squares (T0), whereas taxonomic families are indicated as red vectors. The magnitude of red vectors indicates the strength of their contributions to each PC axis. Vectors pointing in similar directions indicate positively correlated variables, vectors pointing in opposite directions indicate negatively correlated variables. The environmental variables were fitted as blue vectors (dashed lines) onto PC using envfit function from stats R package by calculating multiple regression with $R^2$ scores and *p*-values (top right of the Figure).

Our findings also indicated that there was no difference in the degradation of organic matter between HP and ATM conditions (Figure 2A), as well as there was no significant impact of the TCHO-C/TOC and PCa/Susp. matter in the microbial composition of HP and ATM conditions (Figure 5). Thus, despite a reduction in the phylogenetic diversity in HP samples and a reshaping of the HP community structure, the functional capability of the HP community for degrading organic matter remains undisturbed (i.e., functional resilience). This result is supported by the fact that phyla, families, and genera are absent in the HP community but present in T6-ATM and/or T0 correspond to very low abundant taxa (<1% and mostly <0.5%), which are probably not major contributors to organic matter degradation.

These results support the hypothesis that the pressure increase alone impacts the diversity of active prokaryotes during particle sinking throughout the water column. *Flavobacteriaceae* and *Rhodobacteraceae* are known to be key particle colonizers in surface waters, partly because of their ability to degrade complex substances [79,80]. Bacteroidetes, in particular, displayed many adaptation strategies to grow in a particle-attached lifestyle, such as a wide variety of genes coding for an integrated regulation of adhesion and polymers degradation processes [81]. In a study carried out by Elifantz et al. [82], in coastal water, it has been hypothesized that *Rhodobacteraceae* were pioneer colonizers and Bacteroidetes were secondary colonizers adapted to semi-labile substances. Finding them as belonging to the most active families in our experiment was thus in line with their known ecological traits. Interestingly, *Colwelliaceae* (Gammaproteobacteria), especially the *Colwellia* genus, were among the lowest active families at T0 (0.3% on average) and reached 1.7% at T6-HP. The genus *Colwellia* is known to be one of the few microbial taxa

from which members adapted to life at high hydrostatic pressure have been isolated (see for example [83] and references therein). This bathytype is an opportunistic heterotroph, capable of fast growing in nutrient-rich environments, such as classical culturing media, or when associated with sinking particles. Indeed, *Colwellia* has been found as the second most abundant genus (DNA-level) and one of the most transcriptionally active (RNA-based) inhabitants of the microbiote associated with sinking particles at abyssal depth in the North Pacific Subtropical Gyre [84]. Moreover, Wannicke et al. [85] showed that the piezotolerant strain *Colwellia maris* was able to adapt to the increasing pressure by rising its proportion of unsaturated fatty acids in the membrane phospholipids. This characteristic is recognized in piezophilic strains to optimize membrane fluidity at high pressure [86].

## 4. Conclusions

Using our unique experimental setup and approach, we demonstrated a drastic difference in the composition of transcriptionally active prokaryotes associated with *E. huxleyi*-derived particles comparing incubations performed at atmospheric and at increasing hydrostatic pressure. Hence, we highlighted the necessity to simulate the increase in pressure to which prokaryotes are subjected during the gravitational sinking of particles to accurately characterize microbial key players in particle degradation in the natural environment. Together with previous particle-degradation experiments [34–36] (Table 1) and field experiments [15], this study sheds light on the urgent need to combine geochemical and molecular biology approaches into laboratory simulation to disentangle the apparent imbalance between prokaryotic carbon demand and sinking flux within the deep sea [87]. Going further by reducing the "bottle effect" [88,89] and by including omics approaches such as those carried out by Pelve et al. [90] or Boeuf et al. [84] are promising ways to refine our understanding of processes at play on sinking particles. Such coupled approaches will provide relevant information on the biological actors and key microbial functions or metabolic pathways involved in the degradation of sinking particles in the deep ocean.

**Author Contributions:** C.T. and M.G. designed the experiments; M.G., C.T., V.R., N.B. and S.G. conducted fieldwork; M.G., V.R., S.V., A.B., C.S. and S.G. conducted laboratory analysis; C.T., M.G., P.B. and F.A. analyzed the data. F.A. and M.G. carried out bioinformatics analyses. C.T., M.G., F.A. and D.B. wrote the manuscript. F.A.C.L.M., S.J., P.B., C.P. and C.S. provided critical ideas for interpreting the pathway distribution data and wrote the manuscript. All authors have read and agreed to the published version of the manuscript.

**Funding:** This work was partly funded by Labex OT-Med (ANR-11-LABEX-0061 www.otmed.fr, accessed on 17 September 2021) Investissements d'Avenir, French National Research Agency (ANR, www.agence-nationale-recherche.fr, accessed on 17 September 2021), through the A*Midex ROBIN project (ANR-11-IDEX-0001-684 02 to F.A.C.L.M. and C.T.) with which V.R. was supported. Part of the equipment used in this work was funded by European Regional Development Fund (ERDF; project no 1166-39417).

**Institutional Review Board Statement:** Not applicable.

**Informed Consent Statement:** Not applicable.

**Data Availability Statement:** The raw data are available under the NCBI SRA project PRJNA731017.

**Acknowledgments:** We thank the crew and officers of the R.R.S. DISCOVERY (NERC) for their help during the DY032 cruise. We would like to thank Richard Lampitt (NOC Southampton) for supporting our work at the Porcupine Abyssal Plain. This work is a contribution to the Mediterranean Institute of Oceanography "Pompe Biologique" (BioPump Group). The authors also thank PRECYM, OMICS, SAM, PACEM, and CULTURE—MIO platforms for facilities.

**Conflicts of Interest:** The authors declare no conflict of interest.

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
