# Peer review of "Increasing Hydrostatic Pressure Impacts the Prokaryotic Diversity during Emiliania huxleyi Aggregates Degradation"

_water, doi:10.3390/w13192616_

Round 1
Reviewer 1 Report
In General
This paper describes the influences of hydrostatic pressure and temperature on the prokaryotic diversity associated with the sinking particle derived from E. huxleyi aggregates using a unique apparatus. Information obtained in the present study is very important on the aspect of “Microbial Oceanography”.
However, Discussion on results was unmatured. The authors cited many papers but they often only listed up references and no effective analyses (discussion) on the results obtained were carried out. Preparation of figures and tables are also insufficient.
A reviewer demands the authors to add the major revision on MS after the following comments.
- There are three data sets: T0, T6-HP, and T6-ATM. Differences between T0 and T6-HP are due to organic matter concentration (E. huxleyi aggregates), temperature, and hydrostatic pressure, those between T0 and T6-ATM are due to organic matter and temperature, and difference between T6-HP and T6-ATM is due to hydrostatic pressure.
Therefore, the authors should show clearly throughout the discussion what factor(s) the difference in results was caused by.
- Several previous papers described the reduction of microbial activities and dissolution of carbonate along with increasing in hydrostatic pressure. However, in this study, no such differences were obtained (Fig. 2). What is the difference between the two? A reviewer thought discussion on this point was not enough.
Specific Comments
- M & M, p. 3, line 123 and others
Please give the size of GF/F filters (25 mm or 47 mm in diameter?).
- M & M, p. 5, line 182-184.
How did you remove the weight of salt (desalinization) when you measure the weight of SS? Because weight of salt absorbed by filter paper is not negligible, please give such information in the text.
- M & M, p. 5, line 190-191
What is the difference between LNA and HNA? What microbial communities do you mean by LNA and HNA?
- M & M, p. 6, line 218
Subtitle of 2.4. is same as 2.3. Is it really correct?
- M & M, p. 6, line 227
Table 2 is coming before Table 1.
- M & M, p. 6, line 233 and 240
Give the full text of SD in the line 233, instead of the line 240.
- Results, p. 6, line 262
Why did you select carbohydrate (not protein or others) as the representative parameter of organic matter? Please put some comment on this point.
- Discussion, p. 7, line 265-275
As already mentioned before (in General), although a previous paper [34] described the concentration of particulate carbohydrate decreased more slowly under HP than ATM, no such differences were observed in this study (Fig. 2). What is the difference between the two? Moreover, discussion on the difference between two cited papers ([36] and [66]) is not necessary, only that on your results!
- Fig. 2, p. 8
Put T0, T6-HP, and T6-ATM below the B), too.
Use ATM instead of DEC for the blue mark in C).
As commented in the No. 3, what does increase in the abundance of HNA mean? Please put some discussion on this point.
A reviewer wonders why O2 consumption of ATM increased after 100 h and showed the inverted result. Please put some discussion on this point.
- Discussion, p. 10, line 320-325.
Because references of [34-36] did not change the water temperature and not simulate the in situ condition, comparison of present results and these references is no meaning.
- Discussion, p. 10, line 326-331.
Because, in the present study, activities of bacteria attached to particle were not measured separately from those of free-living, this discussion has no meaning.
- Results, p. 10, line 344, subtitle of 3.3.
[experiments] should be omitted.
- Discussion, p. 10, line 360-361
Please describe more clearly that decrease of phylogenic diversity was due to the effect of water pressure and not by degradation process of organic matter.
- Legend of Table 2
What does [The column of Table 1] in the first line 363 mean?
Column title of [Observed] should be [global richness], which corresponds to the line 351-352.
- Fig. 4, p. 13
A reviewer is afraid this figure lacks some necessary explanations. Please check it more carefully.
- Results, p. 14, line 441 and 446
PC1 and PC2 should be PCA1 and PCA2 (corresponding to the figure).
- Discussion, p. 14, line 451-453 and p. 15, line 493-496
This is the most interesting point of the present study. A reviewer suggests the authors to emphasize this point more.

Reviewer 2 Report
The presented manuscript deals with the influence of hydrostatic pressure on the degradation of bacterial aggregates. It is based on experiments carried out in situ in studies on the existing populations in the deep marine layers. It is an interesting contribution to the knowledge of bacterial growth in rare conditions, as they are not the usual ones in the human habitat.
My only consideration is whether this paper is within the scope of the journal Water. If not it would be more appropriate in another title, such as Microorganisms.
The abstract is correct.
The introduction is well presented and detailed. I feel that it could be better detailed at the end what are the original achievements and contributions of this work and clearly state the objectives of the work. Paragraph 112-116 indicates this but it is not clear to the reader.
The methodology is described perfectly, without problems.
As for the results, each of the results is presented in combination with the discussion of the results, well presented in general. Some details that could be touched up are:
Table 1 contains relevant information, but the font size is small, perhaps it would be convenient to present it on a single page with a larger size horizontally.
In figure 3, the use of colors could be combined with the raster to facilitate the identification of some taxa more clearly. For example, in the case of two similar colors, one could be in plain color and the other forming a raster, to distinguish it better.
As for the conclusions, it would not be necessary to mention bibliographical references. In my opinion it is sufficient to describe the basic sections and the conclusions obtained without including citations or authors again unless it is very necessary. Specifically, paragraph 530-536 includes new citations for the work, therefore, this information should be at the end of the results, before the conclusions and in the conclusions section simply refer to this information.
The bibliography is correct, it is well presented in the style of the journal.
The use of italics in the document should be revised, as it is missing in some taxa and is superfluous in other places, for example in line 31.
Liters is confused with liters (line 124).
Units should be separated from numerical figures, for example it should be 10 m not 10m (line 134 and others).
Reviewer 3 Report
General comments
The manuscript should be proofread for language.
The abstract, as a standalone piece, should provide information on the methods and key findings. The author only mention PASS but do not indicate how the geochemical and microbiological parameters were measured.
Specific comments
Line 13: … demand and the..
Line 30: Delete “incubations”…….
Line 30: … increase in the..
Line 32: … known to be adapted to pressure. Altogether…
Line 84-85: Change “[see 84 details in 34]” to “[34]. Please do the same in other instances within the manuscript.
Line 152-153: Change “(see the plot in Figure 1).” To (Figure 1).
Line 337: occurred
Round 2
Reviewer 1 Report
Reviewer #1
The authors have already revised MS and/or given additional explanations to the reviewer’s comments. The revised MS is almost satisfied to be accepted, except for some points. Please check the following points and revise MS.
Specific Comments (Numbers in previous comments)
- M & M, p. 5, line 182-184. How did you remove the weight of salt (desalinization) when you measure the weight of SS? Because weight of salt absorbed by filter paper is not negligible, please give such information in the text.
→ First: In our sampling protocol, filters are rinsed with few ml of water MQ in order to remove major sea-salt deposit (to prevent icp-ms matrix troubles!) Second: The presence of remaining sea salt was checked by analyzing Na and the sea-salt particulate Ca (and other elements analysed) contribution was found to be negligible.
→OK. Please put such explanation in the text (line 195)
- Legend of Table 2 What does [The column of Table 1] in the first line 363 mean?
→ We are not sure to understand this remark. The column titles are explained in the legend. Input is the raw read and then Filtered, Denoised, Merged and Non-chimeric as common names describing the different steps of the preprocessing of data quality trimming that led to remove bad quality read, chimeric reads. Merged are assembled read. However, we modified the text of the Table 1 caption as followed: Data quality trimming and alpha diversity indices. The column headings of Table 1 indicate the number of reads retained after each processing step of dada2 from the initial raw data (Input). “Filtered” corresponds to the removal of bad quality reads, “Merged” are assembled reads. The “Final retained” column is the read proportion conserved for downstream analyses. The Observed, Shannon, Chao1 and PD (Phylogenetic Diversity) columns provide alpha diversity indices calculated for each sample.
Column title of [Observed] should be [global richness], which corresponds to the line 351-352.
→ ‘Observed’ has been changed by ‘Global richness’ as suggested.
→Table 2, caption.
line 380. [The column of Table 1] must be [The column of Table 2]!
line 381. [dada2] must be [DADA2].
line 383. [Observed] must be [Global richness].
The title of the second column must be [ASV], instead of [Chao1].
- Fig. 4, p. 13 A reviewer is afraid this figure lacks some necessary explanations. Please check it more carefully.
→ The Figure 4 legend has been modified (see lines 470-478) to add better explanation as followed:
Heatmap visualizing the Z-score distribution from the relative abundance of the dominant families (>0.5% in average) initially (T0) and after 6 days of incubation under increasing pressure (T6-HP) and atmospheric pressure (T6-ATM) conditions. Z-score is the number of standard deviations separating a raw score from the mean. For each sample we calculated how far from the mean (of a family relative abundance considering all samples) is the family abundance score of the sample. A positive Z-score indicates the raw score is higher than the mean (red color) and a negative Z-score reveals the raw score is below the mean (blue color). The higher the Z-score, the more overrepresented the family is for the sample compared to the mean. The dendrogram clusters were calculated with UPGMA method (unweighted pair group method with arithmetic mean) according to the Bray-Curtis similarity index. The boxplots in the right panel shows the distribution of family relative abundances (in %) for all samples.
→Our view of Fig. 4 is like this.
Probably upper part and others of Fig. 4 are lacking when giving in the PDF file. Please show the correct figure.

Author Response
Answers to Reviewer #1, Round 2.
The authors have already revised MS and/or given additional explanations to the reviewer’s comments. The revised MS is almost satisfied to be accepted, except for some points. Please check the following points and revise MS.
Specific Comments (Numbers in previous comments)
- M & M, p. 5, line 182-184. How did you remove the weight of salt (desalinization) when you measure the weight of SS? Because weight of salt absorbed by filter paper is not negligible, please give such information in the text.
→ First: In our sampling protocol, filters are rinsed with few ml of water MQ in order to remove major sea-salt deposit (to prevent icp-ms matrix troubles!) Second: The presence of remaining sea salt was checked by analyzing Na and the sea-salt particulate Ca (and other elements analysed) contribution was found to be negligible.
→OK. Please put such explanation in the text (line 195)
ïƒ This sentence has been added as suggested: “The presence of remaining sea salt was checked by analyzing Na and the sea-salt par-ticulate Ca (and other elements analyzed). Their contribution was found to be negligible.”
- Legend of Table 2 What does [The column of Table 1] in the first line 363 mean?
→ We are not sure to understand this remark. The column titles are explained in the legend. Input is the raw read and then Filtered, Denoised, Merged and Non-chimeric as common names describing the different steps of the preprocessing of data quality trimming that led to remove bad quality read, chimeric reads. Merged are assembled read. However, we modified the text of the Table 1 caption as followed: Data quality trimming and alpha diversity indices. The column headings of Table 1 indicate the number of reads retained after each processing step of dada2 from the initial raw data (Input). “Filtered” corresponds to the removal of bad quality reads, “Merged” are assembled reads. The “Final retained” column is the read proportion conserved for downstream analyses. The Observed, Shannon, Chao1 and PD (Phylogenetic Diversity) columns provide alpha diversity indices calculated for each sample.
Column title of [Observed] should be [global richness], which corresponds to the line 351-352.
→ ‘Observed’ has been changed by ‘Global richness’ as suggested.
→Table 2, caption.
line 380. [The column of Table 1] must be [The column of Table 2]!
ïƒ done.
line 381. [dada2] must be [DADA2].
ïƒ done.
line 383. [Observed] must be [Global richness].
ïƒ done.
The title of the second column must be [ASV], instead of [Chao1].
ïƒ As ACE, Chao1 is an estimator of the “true” global richness of a sample (sequencing to the infinity) that can be compared to the number of observed ASV in the same sample (i.e global richness). Chao1 cannot be renamed as “ASV”, otherwise the reader will be confused between the column “Global richness” (which corresponds to number of Observed ASV) and the column “ASV” (which has not really meaning). The reader must be able to distinguish the Observed ASV (Global richness) vs. the estimate of true global richness, and this commonly called Chao1 in published articles.
The Figure 2 caption is now:
“Figure 2: Data quality trimming and alpha diversity indices. The columns headings of Table 2 indicate the number of reads retained after each processing step of DADA2 from the initial raw data (Input). “Filtered” column corresponds to the removal of bad quality reads, “Merged” are assembled reads. The “Final retained” column is the read proportion conserved for downstream analyses. The Global richness (which corresponds to the number of observed Amplicon Sequence Variants – ASV), Shannon, Chao1 and PD (Phylogenetic Diversity) columns provide alpha diversity indices calculated for each sample.”
- Fig. 4, p. 13 A reviewer is afraid this figure lacks some necessary explanations. Please check it more carefully.
→ The Figure 4 legend has been modified (see lines 470-478) to add better explanation as followed:
Heatmap visualizing the Z-score distribution from the relative abundance of the dominant families (>0.5% in average) initially (T0) and after 6 days of incubation under increasing pressure (T6-HP) and atmospheric pressure (T6-ATM) conditions. Z-score is the number of standard deviations separating a raw score from the mean. For each sample we calculated how far from the mean (of a family relative abundance considering all samples) is the family abundance score of the sample. A positive Z-score indicates the raw score is higher than the mean (red color) and a negative Z-score reveals the raw score is below the mean (blue color). The higher the Z-score, the more overrepresented the family is for the sample compared to the mean. The dendrogram clusters were calculated with UPGMA method (unweighted pair group method with arithmetic mean) according to the Bray-Curtis similarity index. The boxplots in the right panel shows the distribution of family relative abundances (in %) for all samples.
→Our view of Fig. 4 is like this.
|
|
|
|
|
|
Probably upper part and others of Fig. 4 are lacking when giving in the PDF file. Please show the correct figure.
ïƒ Sorry for that. The PDF file will be double checked.